# LiDP: Lite Degree Profile Matching with Selective Computations for Random Graph Matching

## Abstract

Random graph matching aims to recover the vertex correspondences between correlated random graphs, but the popular degree profile-based matching method faces prohibitive computational costs in large-scale settings and becomes less effective under high-noise. To address these challenges, we propose an accelerated approach that prioritizes evaluating node pairs in ascending order of the degree difference, rather than utilizing exhaustive distance computations. Theoretical analysis validates that our method reduces complexity from $\tilde{O}(n^2 + nd^2)$ to $\tilde{O}(nd^2)$ while maintaining the guarantee of perfect matching under low-noise. Furthermore, for high-noise scenarios, we extend the approach with a momentum-based acceleration scheme that adaptively adjusts step sizes to ensure convergence. Experiments demonstrate that our method preserves perfect recovery capabilities while achieving superior accuracy and over a $10\times$ speedup compared to the original algorithm.

## 1. Introduction

Graph matching, which is also referred to as network alignment (Feizi et al., 2019), aims to find correspondences between graphs with potential relationships. It can be used in various fields of intelligent information processing, e.g., activity analysis (Chen & Grauman, 2012), shape matching (Michel et al., 2011), detection of similar pictures (Shen et al., 2020), graph similarity computation (Lan et al., 2022), medical image (MH Nguyen et al., 2024), knowledge graph alignment (Xu et al., 2019c), autonomous driving (Song et al., 2023), alignment of vision-language models (Nguyen et al., 2024), and COVID-19 disease mechanism study (Gordon et al., 2020). Graph matching between adjacency matrices $A$ and $B$ can be formulated as the quadratic assignment problem (QAP), defined as:

$$\max_{P \in \Pi_{n \times n}} \mathcal{Z}(P) := \max_{P \in \Pi_{n \times n}} \langle A, PBP^\top \rangle, \qquad (1)$$

with $\Pi_{n \times n}$ denoting the permutation matrix set and $\langle \cdot, \cdot \rangle$ the Frobenius inner product. Although QAP is NP-hard and inapproximable within $2^{\log^{1-\epsilon}(n)}$ for $\epsilon > 0$ in the worst case (Makarychev et al., 2014), these theoretical limits often vary from practical scenarios. Real-world networks typically exhibit latent stochastic structures rather than adversarial worst-case patterns. This distinction has motivated a shift in recent scholarship towards high-probability recovery of the perfect matching (Cullina & Kiyavash, 2017; Lyzinski et al., 2014) rather than worst-case optimization.

Among existing approaches, a profile-based matching method has attracted considerable attention. Ding et al. (2021) developed an $\tilde{O}(nd^2 + n^2)$[1]-time algorithm that perfectly recovers the true vertex correspondence with high probability, provided that the average degree satisfies $d = \Omega(\log^2 n)^2$ and the two graphs differ by at most a $\delta = O(\log^{-2} n)$ fraction of edges. In higher-noise regimes, the authors further introduce a continuous linear assignment–based procedure, referred to as *clean-up*, to refine the matching matrix produced by the profile-based method, which leads to empirical improvements. Still, the resulting computational complexity may limit the applicability of the algorithm to large-scale problems. In addition, the clean-up stage lacks theoretical convergence guarantees.

To address these two issues, we propose the following improvements:

1. **Reduced computation in low-noise regimes.** We compute profile distances in ascending order of node degree differences and terminate the procedure immediately once a perfect matching is identified. This strategy substantially reduces the number of profile distance evaluations and eliminates the $\tilde{O}(n^2)$ sorting step required in the original algorithm, thereby reducing the overall complexity to $\tilde{O}(nd^2)$. Empirically,

[1]Anonymous Institution, Anonymous City, Anonymous Region, Anonymous Country. Correspondence to: Anonymous Author <anon.email@domain.com>.

Preliminary work. Under review by the International Conference on Machine Learning (ICML). Do not distribute.

---

[1]We use notations $\tilde{o}$ and $\tilde{O}$ to hide logarithmic factors.

[2]$a_n = \Omega(b_n)$ and $b_n = O(a_n)$ (or $a_n \gtrsim b_n$ and $b_n \lesssim a_n$) if exist constant $c > 0$ such that $a_n/b_n \geq c$.

we observe that computing only about 5% of all profile distances is sufficient to recover the exact solution, resulting in a 10–20× speedup in total runtime.

2. **Convergence guaranteed clean-up in high-noise regimes.** In the clean-up stage, we relax the original optimization formulation to allow the incorporation of information obtained from the previous step when updating the solution. Moreover, we introduce an adaptive step-size scheme that guarantees convergence of the algorithm. Experimental results demonstrate that this modification improves both accuracy and computational efficiency.

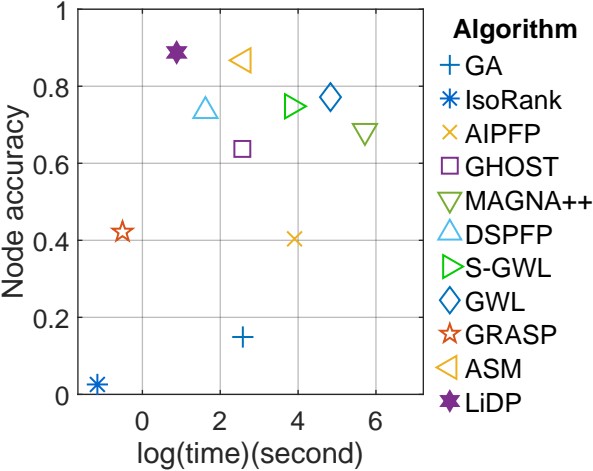

*Figure 1.* Mean matching accuracy and running time of different algorithms on protein network matching (5% noise level).

## 2. Related Work

**Graph Matching** The graph matching paradigm has been applied across a diverse range of fields. In network privacy, it was shown that hidden vertex identities in an anonymized network could be recovered by matching the network to a public one with known identities (Narayanan & Shmatikov, 2008). In systems biology, graph matching is used to discover protein functions by aligning protein-protein interaction networks across different species (Ma & Liao, 2020; Singh et al., 2008). In computer vision, graphs are widely used to represent images, with vertices as image regions and edges encoding adjacency; graph matching is then widely applied to find similar images (Conte et al., 2004). In natural language processing, graph matching is used for tasks like question answering and machine translation by representing sentences as graphs of phrases and their relationships (Kohail & Biemann, 2017; Sun et al., 2020).

**Degree Profile** Many contemporary linear approximation methods for graph matching quantify node similarity us-

ing features derived from the adjacency matrix, such as the degree of each node. However, these features are often highly sensitive to noise, which is a significant drawback in modern applications involving random graph matching. To address this limitation, Ding et al. (2021) proposed an approach that utilizes the second term of the iterated degree sequence (Scheinerman & Ullman, 2011), known as the *degree profile*, as a robust matching feature. This method improves the noise tolerance threshold to $O(\log^{-2} n)$, surpassing the $\tilde{o}(q^2)$ bound of standard degree matching, provided that the average degree satisfies $d = nq \gtrsim \log^2 n$. This demonstrates that the degree profile possesses significantly stronger robustness than simple node degrees when characterizing nodes in noisy environments.

## 3. Preliminary

### 3.1. Erdős-Rényi Graph Model

The statistical analysis of graph matching has recently emerged as a significant area of research, driven by applications in social networks and biology (Pedarsani & Grossglauser, 2011; Barak et al., 2019). Recent work typically assumes that the two graphs, $A$ and $B$, are generated from a random model. The simplest such framework is the correlated Erdős-Rényi graph model:

**Definition 3.1 (Correlated Erdős-Rényi Model $\mathcal{G}(n, q, s)$)**
*Let $n$ be the number of vertices, and let $q, s \in [0, 1]$ be probability parameters. Let $A$ and $B$ denote the adjacency matrices of a pair of correlated random graphs $\mathcal{G}(n, q)$ with the same vertex $[n]$. Denote a permutation $\pi^* : [n] \to [n]$. Conditioned on the realization of $A$, then $B_{\pi^*(i)\pi^*(j)}$ are independent and have the distribution*

$$B_{\pi^*(i)\pi^*(j)} \sim \begin{cases} Binom(1, s) & \text{if } A_{ij} = 1, \\ Binom\left(1, \frac{q(1-s)}{1-q}\right) & \text{if } A_{ij} \neq 1, \end{cases} \quad (2)$$

*where $Binom(k, p)$ represents the binomial distribution with $k$ trials and success probability $p$.*

### 3.2. Degree Profile Matching

Ding et al. (2021) propose a random graph matching method based on *degree profiles* to recover vertex correspondences. The degree profile of a vertex is defined as the empirical distribution of the degrees of its neighbors and is treated as a structural signature of the vertex. The key idea is that, for a truly matched vertex pair, the correlation between the two graphs induces a large number of corresponding common neighbors, resulting in similar degree profiles. In contrast, for an incorrectly matched pair, the neighborhoods are largely independent, and the corresponding degree profiles differ significantly in distribution. By discretizing degree profiles and measuring the similarity between vertex pairs

using the total variation ($\ell_1$) distance, the graph matching problem is reduced to a sorting task.

**Degree Profile.** Consider a graph $A$ on $n$ vertices. For any vertex $i \in [n]$, let $N_A(i) = \{j : A_{ij} = 1\}$ denote its open neighborhood and $a_i = |N_A(i)|$ denote its degree. The closed neighborhood is defined as $N_A[i] = N_A(i) \cup \{i\}$. To construct the degree profile, we first define the standardized degree of a neighbor $j \in N_A(i)$ with respect to $i$ as

$$a_j^{(i)} \triangleq \frac{1}{\sqrt{(n - a_i - 1)q(1-q)}} \sum_{l \notin N_A[i]} (A_{lj} - q), \quad (3)$$

where $q$ is the connection probability. Based on these values, the empirical measure $\mu_i$ for vertex $i$ is defined as

$$\mu_i \triangleq \frac{1}{a_i} \sum_{j \in N_A(i)} \delta_{a_j^{(i)}}, \quad (4)$$

where $\delta_x$ represents the Dirac measure centered at $x$. Finally, the degree profile $\bar{\mu}_i$ is obtained by centering $\mu_i$ with its theoretical distribution

$$\bar{\mu}_i = \mu_i - \overline{\mathrm{Binom}}(n - a_i - 1, q), \quad (5)$$

where $\overline{\mathrm{Binom}}$ denotes the law of the standardized binomial distribution. The degree profile $\bar{\nu}_i$ for graph $B$ is defined analogously.

**Degree Profile Distance.** To quantify the difference between degree profiles and enable vertex matching, we compare the degree profile measures via a discretized $\ell_1$ distance. Specifically, we fix an integer $L \in \mathbb{N}$ and construct a uniform partition $I_1, \ldots, I_L$ of the interval $[-1/2, 1/2]$, where each sub-interval has width $|I_l| = 1/L$. For any pair of vertices $(i, j)$ from the two graphs, we define the degree profile distance as

$$D_{ij} = \sum_{l \in [L]} |\bar{\mu}_i(I_l) - \bar{\nu}_j(I_l)| := \|\bar{\mu}_i - \bar{\nu}_j\|_{1,L}. \quad (6)$$

The degree profile distance serves as a matching score for vertex pairs: Truly corresponding vertices tend to have small distances due to correlated local structures, whereas incorrectly matched pairs typically yield much larger values.

**Degree Profile Matching Algorithm.** Building upon the resulting distance matrix $D$ defined in equation (6), where $D_{ij}$ measures the structural dissimilarity between vertex $i$ in graph $A$ and vertex $j$ in graph $B$, Ding et al. (2021) propose a greedy matching strategy. Specifically, the algorithm sorts all $n^2$ entries of $D$ and retains the $n$ smallest ones. If these candidates form a valid permutation, they are accepted as the recovered vertex correspondence. This procedure is justified by the strong statistical separation between the distances of true matches and those of incorrect pairs in correlated random graphs.

---

**Algorithm 1** Graph matching via degree profiles (DP Algorithm, (Ding et al., 2021))

---

**Require:** Graphs $A$ and $B$ on $n$ vertices.
**Ensure:** A permutation $\hat{\pi}$ of $n$ elements.
1: Compute profile distance $D_{ik}$ between each $i, k \in [n]$.
2: Sort $\{D_{ik} : i, k \in [n]\}$ and let $\mathcal{S}$ be the set of indices of the smallest $n$ elements.
3: **if** $\mathcal{S} = \{(i, \hat{\pi}(i)) : i \in [n]\}$ form some permutation $\hat{\pi}$ **then**
4:     **return** $\hat{\pi}$.
5: **else**
6:     **return** Error.
7: **end if**

---

### 3.3. Clean-up Strategy

Since the theoretical guarantees of Algorithm 1 are restricted to limited noise regimes, Ding et al. (2021) employed an iterative clean-up strategy to enhance robustness in high-noise environments. Initialized with the permutation matrix from Algorithm 1, this procedure refines the matching by iteratively solving a linear assignment problem,

$$P^0 = D, \quad P^{t+1} = \arg \max_{P \in \Pi_{n \times n}} \langle P, AP^t B \rangle, \quad (7)$$

which aims to maximize the number of common neighbors consistent with the current alignment.

## 4. Lite Degree Profile Matching

### 4.1. Selective Computation Scheme

We observe that matches typically occur between nodes with similar degrees. This implies that it may be sufficient to identify the smallest $n$ degree profile distances in the procedure of computing the entire matrix $D$. A natural acceleration strategy, therefore, is to compute the degree profile distances between node pairs as the ascending order, aiming to obtain an optimal match quicker. The core challenge of this method lies in determining–without computing the complete matrix–when the smallest $n$ elements have been reliably identified. The whole procedure is detailed in Figure 2.

After computing the degree profile distances for all node pairs whose degree difference does not exceed $d$, we denote the $n$ smallest among them as $\mathcal{S}_n^{(d)}$. To determine whether the corresponding elements $D_{ij}$ in $\mathcal{S}_n^{(d)}$ are the globally smallest $n$ elements, we rely on the following theorem: a set of $n$ elements can be the globally smallest only if each of them is the minimum value in both its row and its column of the complete degree profile distance matrix.

**Theorem 4.1** *Let $D$ be an $n \times n$ real matrix. If the positions of the $n$ smallest elements of $D$ form a permutation matrix,*

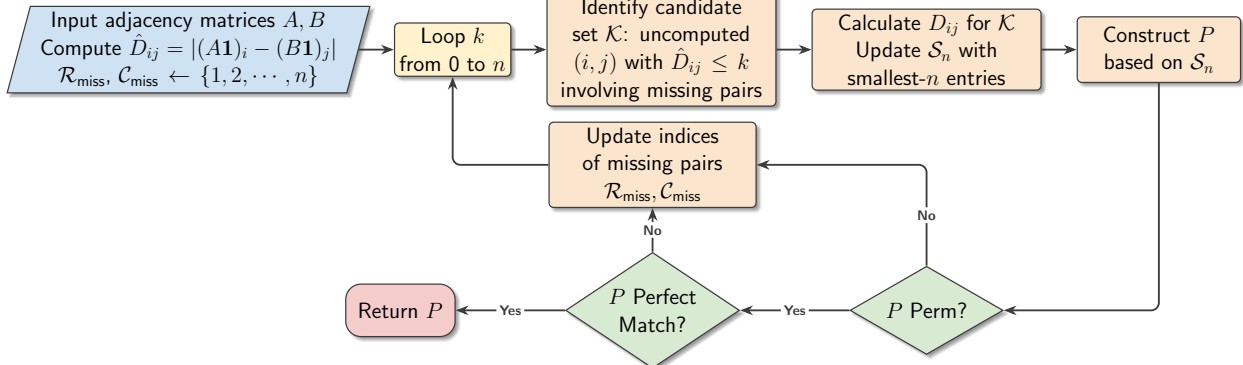

*Figure 2.* The exact recovery procedure of the LiDP algorithm. $\mathcal{R}_{\text{miss}}$ and $\mathcal{C}_{\text{miss}}$ represent the row and column indices from $\{1, \ldots, n\}$ that are not yet present in the current construction of $P$.

*then each of these elements is the minimum in its row and the minimum in its column.*

**Proof** *Let $S$ be the set of positions of the $n$ smallest entries of $D$. Fix any $(i, j) \in S$.*

***Step 1:** $D_{ij}$ is the minimum in its $i$-th row. Suppose, for the sake of contradiction, that there exists some $j' \neq j$ such that $D_{ij'} < D_{ij}$. Since $D_{ij'}$ is strictly smaller than $D_{ij}$ and $S$ consists of the $n$ smallest entries of $D$, the position $(i, j')$ must also be included in $S$. This contradicts the assumption that $S$ contains exactly one position in each row. Hence $D_{ij}$ must be the minimum entry in its $i$-th row.*

***Step 2:** $D_{ij}$ is the minimum in its $j$-th column. A symmetric argument applies to columns. If there exists $i' \neq i$ such that $D_{i'j} < D_{ij}$, then the position $(i', j)$ would also belong to $S$, contradicting the assumption that each column contains exactly one position from $S$. Therefore, $D_{ij}$ must also be the minimum entry in its $j$-th column.*

*Combining Steps 1 and 2, we conclude that for every $(i, j) \in S$, the entry $D_{ij}$ is the minimum in both its row and its column.*

Based on Theorem 4.1, we only need to verify whether the largest element in $\mathcal{S}_n^{(d)}$ is the smallest degree profile distance within its corresponding row and column. If this condition holds, then all elements in $\mathcal{S}_n^{(d)}$ are guaranteed to be among the globally smallest $n$ elements. Algorithm 2, along with Figure 3, details the subroutine used for this verification.

Assuming that the degree profile distance matrix restricted to degree differences not exceeding $d$ already contains the final set $\mathcal{S}_n$, it is in fact unnecessary to compute the entire matrix in order to identify $\mathcal{S}_n$. The procedure proceeds as follows. First, we compute the degree profile distances for node pairs with zero degree difference, select the $n$ smallest values to form $\mathcal{S}_n^{(0)}$, and encode them in a binary matrix

**Algorithm 2** Check Perfect

**Require:** Current matching candidates $\mathcal{I}, \mathcal{J}$ and corresponding distance $\mathcal{D}$

**Ensure:** Updated $\mathcal{I}, \mathcal{J}, \mathcal{D}$ and Boolean flag $isPerfect$

1: Initialization $isPerfect \leftarrow$ **True**
2: **for** $k = 1$ **to** $n$ **do**
3:     Compute $d_{row} = D_{r^*,k}$ and $d_{col} = D_{k,c^*}$ by (6)
4:     **if** $d_{row} < \text{last}(\mathcal{D})$ **then**
5:         Remove $c^*$ and $\tau$ from $\mathcal{J}$ and $\mathcal{D}$
6:         Insert $d_{row}$ to $\mathcal{D}$ and maintain order
7:         Insert $k$ to $\mathcal{J}$ corresponding to $\mathcal{D}$
8:         $isPerfect \leftarrow$ **False**
9:     **end if**
10:     **if** $d_{col} < \text{last}(\mathcal{D})$ **then**
11:         Remove $r^*$ and $\tau$ from $\mathcal{I}$ and $\mathcal{D}$
12:         Insert $d_{col}$ to $\mathcal{D}$ and maintain order
13:         Insert $k$ to $\mathcal{I}$ corresponding to $\mathcal{D}$
14:         $isPerfect \leftarrow$ **False**
15:     **end if**
16: **end for**
17: **return** $\mathcal{I}, \mathcal{J}, \mathcal{D}, isPerfect$

$\hat{P}^{(0)}$, where entries corresponding to candidate elements are set to 1. In subsequent iterations (e.g., $d = 1$), we compute only those degree profile distances with degree difference at most $d$ whose rows and columns are not yet covered by $\mathcal{S}_n^{(0)}$ in $\hat{P}^{(0)}$, thereby updating the candidate set. This process is repeated, progressively expanding the scope of computation, until the globally smallest $n$ elements $\mathcal{S}_n$ are obtained. We prove that this strategy does not affect accuracy.

**Theorem 4.2 (Guarantee of LiDP)** *Let $s = 1 - \sigma^2$ and $q \leq q_0$ for some sufficiently small positive constant $q_0$. Assume that*

$$\sigma \leq \frac{\sigma_0}{\log n}, \tag{8}$$

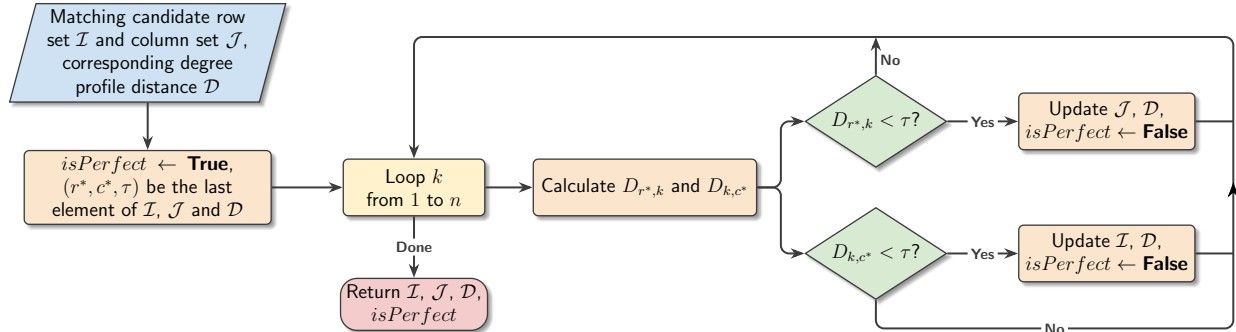

*Figure 3.* Check perfect algorithm description. The input $\mathcal{I}$, $\mathcal{J}$ are the row and column indices of non-zero element in $P$, its corresponding degree profile distance is $\mathcal{D}$.

*for some sufficiently small absolute constant $\sigma_0$. Set*

$$L = L_0 \log n, \tag{9}$$

*and assume that*

$$nq \geq C_0 \log^2 n, \tag{10}$$

*for some large absolute constants $L_0, C_0$. Then with probability $1 - O(1/n)$, the LiDP algorithm (Alg. 4) successfully recovers the perfect matching $\widehat{\pi} = \pi^*$.*

**Proof** *The proof is divided into two parts: the first case is when LiDP finds a permutation matrix and terminates early by passing the perfect test; the second case is when the algorithm runs to completion. In the following analysis, $\mathcal{S}$ denotes the permutation corresponding to Algorithm 1, while $\mathcal{S}_n$ represents the set of candidate matched pairs generated by our LiDP.*

### Case 1: Early termination via the perfect test.

*Passing the perfect test in Figure 2 implies that for*

$$(i^*, j^*) = \arg \max_{(i,j) \in \mathcal{S}_n} D_{ij}, \tag{11}$$

*we have*

$$D_{i^*j^*} < \min\Big\{ \min_{j' \neq j^*} D_{i^*j'}, \ \min_{i' \neq i^*} D_{i'j^*} \Big\}. \tag{12}$$

*Since $\mathcal{S}$ in Algorithm 1 corresponds to the perfect matching, by Theorem 4.1 we have*

$$D_{ij} < \min\Big\{ \min_{j' \neq j} D_{ij'}, \ \min_{i' \neq i} D_{i'j} \Big\} \Leftrightarrow (i,j) \in \mathcal{S}. \tag{13}$$

*Hence, $(i^*, j^*) \in \mathcal{S}$. Moreover, for any other element in $\mathcal{S}_n$,*

$$D_{ij} < D_{i^*j^*}, \quad \forall (i,j) \in \mathcal{S}_n. \tag{14}$$

*Since*

$$|\mathcal{S}| = n, \qquad \forall (i,j) \in \mathcal{S}, \ \forall (a,b) \notin \mathcal{S}, \ D_{ij} < D_{ab}, \tag{15}$$

*we have*

$$\mathcal{S}_n \setminus \{(i^*, j^*)\} \subset \mathcal{S} \ \Rightarrow \ \mathcal{S}_n = \mathcal{S}. \tag{16}$$

### Case 2: Runs to completion.

*Suppose that the solution $P$ found by LiDP differs from the optimal solution $P^*$ found by DP. This implies that $\mathcal{S}_n \neq \mathcal{S}$.*

*Since the 'Check Perfect' step represented in Figure 3 updates any non-perfect permutation matrix into a $0-1$ matrix that is not a permutation matrix, once $\mathcal{S}_n \neq \mathcal{S}$, the matrix $P$ derived from $\mathcal{S}_n$ cannot be a permutation matrix. Hence,*

$$\mathcal{R}_{\text{miss}} \cup \mathcal{C}_{\text{miss}} \neq \varnothing, \tag{17}$$

*therefore,*

$$D_{ij} > D_{ab}, \ \forall i \in \mathcal{R}_{\text{miss}} \ or \ j \in \mathcal{C}_{\text{miss}}, \ (a,b) \in \mathcal{S}_n. \tag{18}$$

*From the definitions of $\mathcal{S}$ and $\mathcal{S}_n$, we have*

$$\max_{(i,j) \in \mathcal{S}_n} D_{ij} \geq \max_{(k,l) \in \mathcal{S}} D_{kl}. \tag{19}$$

*Therefore,*

$$(i,j) \notin \mathcal{S}, \quad \forall i \in \mathcal{R}_{\text{miss}} \ or \ j \in \mathcal{C}_{\text{miss}}. \tag{20}$$

*This implies that $\mathcal{S}$ cannot form a permutation matrix, which contradicts the assumption. Hence,*

$$\mathcal{S}_n = \mathcal{S}. \tag{21}$$

*By applying Theorem 2 of (Ding et al., 2021), the proof is completed.*

**Remark 4.3** *Theorem 4.2 demonstrates that LiDP matches the statistical optimality of the DP algorithm (Ding et al., 2021) under the same noise regimes. Crucially, LiDP achieves these guarantees with substantially lower computational cost. Thus, LiDP offers an algorithmic improvement that enhances efficiency without compromising the probability of exact recovery.*

## 4.2. Momentum-Adaptive Clean-up Procedure

While the selective computation scheme ensures exact recovery with high probability under moderate noise, structural perturbations exceeding the theoretical threshold may prevent immediate convergence to the global optimum. Nevertheless, the resulting approximate matching provides a highly informative warm start. To refine this candidate solution and rectify remaining mismatches, we propose a momentum-based clean-up strategy that incorporates an adaptive step size. Let $P^t$ denote the permutation matrix at iteration $t$. The update rule is defined as:

$$P^{t+1} = P^t + \alpha^t(\tilde{P}^t - P^t), \tag{22}$$

$$\tilde{P}^t = \arg\max_{P \in \Pi_{n \times n}} \langle P, AP^t B \rangle, \tag{23}$$

$$\alpha^t = \arg\max_{\alpha \in [0,1]} \mathcal{Z}(P^t + \alpha(\tilde{P}^t - P^t)). \tag{24}$$

Here, the Frank-Wolfe direction (Polyak & LEVITIN, 1966) $\tilde{P}^t$ is generated based on the current gradient approximation. Equation (23) is solved by the Hungarian algorithm, yielding a permutation matrix $\tilde{P}^t$. According to the convergence analysis in (Shen et al., 2024b), the sequence $\{P^t\}$ generated by the above rules is guaranteed to converge to a local maxima.

**Theorem 4.4** *Let $A, B \in \mathbb{R}^{n \times n}$ be two adjacency matrices. For any arbitrary initialization $D$, the sequence $\{P^{(t)}\}$ generated by Algorithm 3 converges to a local maximum of the objective function (1).*

**Proof** *Let $\Delta P^t = \tilde{P}^t - P^t$. Substituting the update rule into the objective function $\mathcal{Z}(P) = \frac{1}{2}tr(P^\top APB)$, we expand the term with respect to $\alpha$*

$$
\begin{aligned}
\mathcal{Z}(P^t + \alpha\Delta P^t) =& \frac{1}{2}tr\left[(P^t + \alpha\Delta P^t)^\top A(P^t + \alpha\Delta P^t)B\right] \\
=& \frac{1}{2}tr(P^{t\top}AP^t B) + \alpha tr(\Delta P^{t\top}AP^t B) + \\
& \frac{1}{2}\alpha^2 tr(\Delta P^{t\top}A\Delta P^t B) \\
=& c + b\alpha + a\alpha^2,
\end{aligned}
\tag{25}
$$

*where the coefficients are identified as*

$$a = \frac{1}{2}tr(\Delta P^{t\top}A\Delta P^t B), \quad b = tr(\Delta P^{t\top}AP^t B). \tag{26}$$

*Since $\tilde{P}^t = \arg\max_P \langle P, \nabla\mathcal{Z}(P^t)\rangle$, we have*

$$\langle \tilde{P}^t, \nabla\mathcal{Z}(P^t)\rangle \geq \langle P^t, \nabla\mathcal{Z}(P^t)\rangle, \tag{27}$$

*which explicitly implies $b \geq 0$. Therefore, the optimal $\alpha^t$ is determined by*

$$
\alpha^t = \begin{cases} 1, & \text{if } a \geq 0, \\ \min\left(1, -\frac{b}{2a}\right), & \text{if } a < 0. \end{cases}
\tag{28}
$$

*By invoking Property 1 and 2 in (Shen et al., 2024b), we establish that the sequence $\{P^t\}$ converges to a local maximum.*

---

**Algorithm 3** Momentum-Adaptive Clean-up Procedure

---

**Require:** Adjacency matrices $A, B \in \mathbb{R}^{n \times n}$, initial $D$, maximum iterations $T$, tolerance $\epsilon$.
**Ensure:** Permutation matrix $P$.
1: $P^0 \leftarrow \mathcal{P}(-ADB)$
2: $\rho_0 \leftarrow \frac{1}{n}\langle D, P^0 \rangle$
3: **for** $t = 0$ **to** $T$ **do**
4:     $\tilde{P}^t \leftarrow \mathcal{P}(-AP^t B)$
5:     Compute step size $\alpha^t$ via (24)
6:     $P^{t+1} \leftarrow P^t + \alpha^t(\tilde{P}^t - P^t)$
7:     $\rho_{t+1} \leftarrow \frac{1}{n}\langle P^{t+1}, P^t \rangle$    ▷ # Convergence check
8:     **if** $|\rho_{t+1} - \rho_t| < \epsilon$ **then**
9:         **break**
10:     **end if**
11: **end for**
12: $P \leftarrow \mathcal{P}(-AP^{T+1}B)$
13: **return** $P$

---

### 4.3. LiDP Algorithm

Algorithm 4 integrates these two strategies into a unified framework. This process implements a selective computation strategy that allows for early termination as soon as global optimality is met. If verification fails, the algorithm seamlessly transitions to the momentum-adaptive clean-up phase. By leveraging the pre-computed distance matrix as a warm start, LiDP maintains high speed under low precision, while preserving robustness in subsequent iterations.

## 5. Experiments

We evaluate the performance of the proposed LiDP algorithm from the following two aspects

> **Q1.** How does LiDP benchmark against the standard DP algorithm in low-noise regimes?

> **Q2.** How robust is the momentum-accelerated clean-up strategy under higher noise variations?

**Datasets** include synthetic correlated Erdős-Rényi graphs $\mathcal{G}(n, p, 1 - \delta)$ and two real-world datasets: the Slashdot social network and the yeast PPI network. For the real-world cases, we utilize the following as parent graphs: a subgraph induced by the first 750 users from Slashdot, and the high-confidence *S. cerevisiae* PPI network (Collins et al., 2007). The relevant setting details are summarized in Table 1.

**Algorithm 4** Lite Degree Profile Matching Algorithm (LiDP Algorithm)

**Require:** Adjacency matrices $A, B \in \{0,1\}^{n \times n}$.
**Ensure:** Permutation matrix $P$, distance $D$.
1: Compute $\hat{D}_{ij} = |(A\mathbf{1})_i - (B\mathbf{1})_j|$ and generate profiles via (5).
2: Initialize $\mathcal{S}_n \leftarrow \varnothing, \Omega \leftarrow \varnothing$.
3: **for** $k = 0$ **to** $\max(\hat{D})$ **do**
4:  $\mathcal{K} \leftarrow \{(i,j) \mid (i \in \mathcal{R}_{\text{miss}} \vee j \in \mathcal{C}_{\text{miss}}) \wedge (\hat{D}_{ij} \leq k)\} \setminus \Omega$
5:  **for all** $(i,j) \in \mathcal{K}$ **do**
6:    Compute $D_{ij}$ via (6) and add $(i,j)$ to $\Omega$.
7:    Update $\mathcal{S}_n$ to keep the $n$-smallest pairs corresponding to $D$
8:  **end for**
9:  Extract $\mathcal{I}_n$ and $\mathcal{J}_n$ from $\mathcal{S}_n$, and let $\mathcal{D}_n$ be the $n-$smallest elements in $D$
10:  **if** $\mathcal{S}_n$ forms a valid bijection **then**
11:    $(\mathcal{I}_n, \mathcal{J}_n, \mathcal{D}_n) \leftarrow$ Check Perfect$(\mathcal{I}_n, \mathcal{J}_n, \mathcal{D}_n)$
12:    **if** $\mathcal{S}_n$ forms perfect matching **then**
13:      Construct $P$ based on $\mathcal{S}_n$
14:      **return** $P$
15:    **end if**
16:  **end if**
17:  $\mathcal{R}_{\text{miss}} \leftarrow \{1, \cdots, n\} \setminus \mathcal{I}_n, \mathcal{C}_{\text{miss}} \leftarrow \{1, \cdots, n\} \setminus \mathcal{J}_n$
18: **end for**
19: ▷ # *Cannot find perfect matching, use clean-up*
20: $P \leftarrow$ Algorithm 3 with initialization $D$
21: **return** $P$

*Table 1.* Summary of the datasets and experimental settings. The noise level is controlled by the deletion probability $\delta$.

| Dataset | Dims (n) | Noise Level |
|---|---|---|
| Synthetic | [1000, 10000] | $\sqrt{\delta} \in [0, 0.4]$ |
| Slashdot | 750 | $\delta \in [0, 0.4]$ |
| Yeast PPI | 1004 | $\delta \in \{0\%, 5\%, 10\%, 15\%\}$ |

**Baselines** The comparison is conducted in two settings. For the synthetic and Slashdot datasets, we benchmark against the standard DP and DP+ (Ding et al., 2021) to validate the robustness of our clean-up strategy. For the Yeast PPI network, we evaluate performance against comprehensive state-of-the-art methods, including GA (Gold & Rangarajan, 1996), IsoRank (Singh et al., 2008), AIPFP (Leordeanu et al., 2009; Lu et al., 2016), GHOST (Patro & Kingsford, 2012), MAGNA++ (Vijayan et al., 2015), DSPFP (Lu et al., 2016), S-GWL (Xu et al., 2019a), GWL(Xu et al., 2019b), GRASP (Hermanns et al., 2023), and ASM (Shen et al., 2024a).

## 5.1. Perfect Matching Comparison

We evaluate the performance of our LiDP Algorithm (Algorithm 4) combined with Algorithm 2 and DP Algorithm (Algorithm 1) proposed in (Ding et al., 2021) on the correlated Erdős-Rényi graph model $\mathcal{G}(n, q, 1-\delta)$.

Figure 4 demonstrates the substantial computational advantage of LiDP over DP. In scenarios where the standard DP algorithm successfully identifies a perfect matching, our method achieves a remarkable speedup of up to $21\times$. This indicates that LiDP is capable of recovering the ground truth with the same reliability as the full DP approach with only a fraction of the computational cost. Moreover, Table 2 provides a detailed analysis of this efficiency gain.

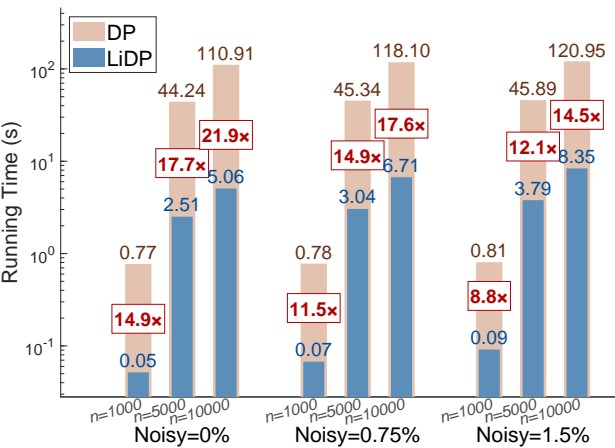

*Figure 4.* Speedup of our LiDP over DP across different $n$ and noise levels on Erdős-Rényi graph, values inside the red boxes represent the speedup ratios.

*Table 2.* Comparison of computed degree profile distances between DP and LiDP on Erdős-Rényi graphs.

| Problem Size | DP | LiDP | | Cost Reduction | |
|---|---|---|---|---|---|
| | | Noisy 0% | Noisy 2% | Noisy 0% | Noisy 2% |
| $n = 1,000$ | $1.00 \times 10^6$ | $4.31 \times 10^4$ | $5.08 \times 10^4$ | 95.69% | 94.92% |
| $n = 5,000$ | $2.50 \times 10^7$ | $8.34 \times 10^5$ | $1.02 \times 10^6$ | 96.66% | 95.92% |
| $n = 10,000$ | $1.00 \times 10^8$ | $3.10 \times 10^6$ | $3.86 \times 10^6$ | 96.90% | 96.14% |

## 5.2. Clean-up Strategy Comparison

As clearly evidenced by the comparative results presented in Figures 5 and 6, the proposed momentum-adaptive clean-up strategy demonstrates significantly superior effectiveness and robustness.

As shown in Figure 5(a), our proposed strategy significantly enhances the robustness of the matching algorithm. Although perfect recovery is no longer guaranteed beyond the theoretical upper bound, LiDP successfully delays the phase transition, maintaining perfect accuracy up to $\sqrt{\delta} \approx 0.4$, while DP+ suffers a sharp performance breakdown at $\sqrt{\delta} \approx 0.25$. This indicates that the momentum-

*Table 3.* Comparisons on yeast PPI with node accuracy(%) and runtime(s) across different noise levels.

| Yeast network | 0% noise | | 5% noise | | 10% noise | | 15% noise | |
|---|---|---|---|---|---|---|---|---|
| Methods | Node Acc | time | Node Acc | time | Node Acc | time | Node Acc | time |
| GA (Gold & Rangarajan, 1996) | 24.0% | 16s | 14.0% | 12.2s | 11.8% | 12.3s | 9.6% | 12.3s |
| IsoRank (Singh et al., 2008) | 7.8% | 0.3s | 2.3% | 0.3s | 0.1% | 0.3s | 0.1% | 0.4s |
| AIPFP (Leordeanu et al., 2009; Lu et al., 2016) | 56.2% | 63.8s | 43.1% | 52.7s | 35.1% | 45.2s | 27.1% | 37.6s |
| GHOST (Patro & Kingsford, 2012) | 86.1% | 10.1s | 74.8% | 12.9s | 56.1% | 14.0s | 39.9% | 15.1s |
| MAGNA++ (Vijayan et al., 2015) | 88.3% | 290.3s | 77.6% | 301.7s | 61.9% | 308.5s | 45.8% | 315.3s |
| DSPFP (Lu et al., 2016) | 85.9% | 4.9s | 78.1% | 5.1s | 69.5% | 5.1s | 60.8% | 5.1s |
| S-GWL (Xu et al., 2019a) | 83.6% | 63.1s | 81.3% | 41.2s | 71.9% | 41.2s | 62.4% | 41.1s |
| GWL(Xu et al., 2019b) | 83.7% | 140.8s | 83.7% | 113.2s | 75.0% | 120.3s | 66.3% | 127.4s |
| GRASP (Hermanns et al., 2023) | 98.3% | 0.4s | 38.6% | **0.6s** | 23.5% | **0.7s** | 8.3% | **0.7s** |
| ASM (Shen et al., 2024a) | 94.0% | 10s | 89.8% | 17.8s | 82.2% | 14.6s | **80.9%** | 11.4s |
| LiDP | **100%** | **0.13s** | **89.9%** | 1.7s | **84.7%** | 3.0s | 80.5% | 4.8s |

adaptive method effectively mitigates the impact of structural noise. A similar trend is observed in Figure 6(a), where the LiDP variants consistently outperform the DP+ across all noise levels, exhibiting a slower decay in accuracy as $\delta$ increases.

Regarding runtime efficiency (Figure 5(b) and 6(b)), the introduction of our adaptive step size strategy induces a moderate increase in computational time. However, this overhead is justifiable given the substantial gains in accuracy. Notably, on the Slashdot network, the runtime gap between the proposed and baseline methods remains narrow, suggesting that the proposed strategy maintains good scalability on practical, real-world networks.

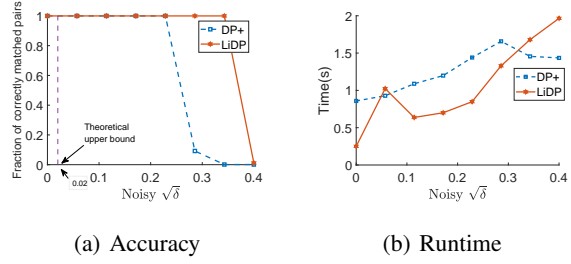

| (a) Accuracy | (b) Runtime |
|---|---|

*Figure 5.* Accuracy and runtime comparison between DP+ and LiDP on Erdős-Rényi graph, the dashed vertical line indicates the theoretical upper bound for noise tolerance.

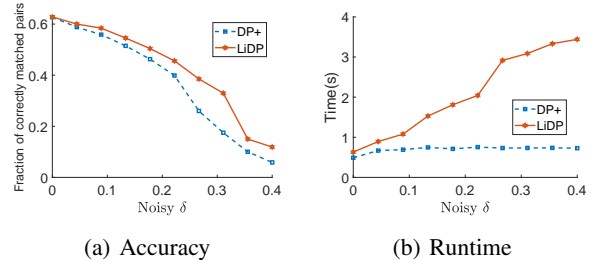

| (a) Accuracy | (b) Runtime |
|---|---|

*Figure 6.* Accuracy and runtime comparison between DP+ and LiDP on Slashdot network.

To assess the performance of the proposed algorithm in practical scenarios, we extend our evaluation to the Yeast PPI matching. Table 3 presents the node accuracy and runtime comparisons between LiDP and many other classic methods under varying noise levels.

# 6. Conclusion

In this paper, we proposed the Lite Degree Profile algorithm (LiDP) and a momentum adaptive clean-up strategy to enhance both the computational efficiency and noise robustness of graph matching. By leveraging a degree-difference pruning mechanism, LiDP reduces the number of computed entries by approximately 97%, achieving up to a $21\times$ speedup on Erdős-Rényi graphs while theoretically maintaining the exact recovery guarantees of standard degree profile methods. Moreover, our momentum-adaptive clean-up scheme provides theoretical convergence guarantees and delays the phase transition in high-noise regimes.

Future work will focus on three main directions. First, we aim to extend the theoretical framework beyond unweighted graphs to general weighted graphs and other random graph models. Second, we plan to refine the adaptive step size selection mechanism to further accelerate the convergence rate of the clean-up phase. Finally, we intend to incorporate matrix-free techniques to reduce memory complexity, enabling the framework to scale effectively to large-scale real-world problem.

## Impact Statement

This paper presents work whose goal is to advance the field of Machine Learning. There are many potential societal consequences of our work, none which we feel must be specifically highlighted here.

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
