# OpenReview forum: "LiDP: Lite Degree Profile Matching with Selective Computation for Random Graph Matching"
_ICML.cc/2026/Conference — Submitted to ICML 2026_

### Official Review · Reviewer_pCxx · 2026-02-14

**Soundness:** 2
**Presentation:** 3
**Significance:** 3
**Originality:** 2
**Overall Recommendation:** 3
**Confidence:** 4

**Summary:**

This paper proposes LiDP, a new method based on degree profile method for graph matching in large scale setting. Specifically, the authors provide theoretical analysis and reduce complexity from \tilde{O}(n^2 + nd^2) to \tilde{O}(nd^2), while maintaining the guarantee of  perfect matching. Also, synthetic data and realdata analysis demostrate that the method performs well on large scale data (n \ge 1000).

**Compliance With Llm Reviewing Policy:**

Affirmed.

**Final Justification:**

I still have concerns despite the authors’ efforts in the rebuttal. In particular, I feel that the paper tends to overstate its contributions. For example, subgraph counting methods can already be applied to graphs with around 1000 nodes using color-coding-based algorithms, whereas the rebuttal seems to suggest otherwise. Given this issue and the similarity to prior work, I remain unconvinced and therefore maintain my negative score.

**Key Questions For Authors:**

The main question is whether the proposed method provides genuinely new methodological or theoretical contributions beyond the classical degree-profile framework (Ding et al., 2021), and in what regimes it offers advantages over existing graph matching approaches. In particular, can the authors clarify the novelty of the modified distance matrix D, and explain under which settings their method improves over subgraph-counting–based or iterative algorithms that already achieve recovery under constant correlation?

**Limitations:**

The proposed approach appears to build heavily on the classical degree-profile framework, with the main change being a modification of the distance construction, and much of the theoretical guarantee seems to follow from existing analyses. In addition, the method does not appear to be state-of-the-art in terms of statistical thresholds, as it typically requires near-perfect correlation 1-o(1), whereas several recent subgraph-counting–based or iterative algorithms succeed under constant correlation in correlated ER graph matching and correlation detection. The empirical evaluation could also be strengthened: comparisons with stronger baselines are missing, and some experimental behaviors—such as the performance beyond the theoretical noise bound in Figure 5a—would benefit from further explanation. Finally, it remains unclear whether the approach extends to more general settings, such as inhomogeneous models (e.g., Ding–Fei–Wang, 2025), which limits the current scope of applicability.

**Strengths And Weaknesses:**

The paper proposes a new method based on the degree-profile approach, and the experiments show that it can be applied to graph matching scenarios with n \ge 1000. I find the paper quite interesting, as many graph matching algorithms with theoretical guarantees are difficult to run on large-scale graphs.

However, I have several concerns. First, most components of the proposed method appear to be very similar to the degree-profile method in Ding et al. (2021), and the authors mainly make small modifications to the distance matrix D. The theoretical guarantees also seem to rely largely on prior work (while the original analysis for the degree-profile method is already highly non-trivial).

Another important concern is that, although the degree-profile method can succeed for graph matching, it is not a state-of-the-art algorithm: it typically succeeds only under 1-o(1) correlation, whereas many subgraph-counting–based or iterative algorithms can succeed under constant correlation in the correlated \mathrm{ER} graph matching problem (or correlation detection problem). It would therefore be helpful if the authors could compare their method in more detail with these approaches (e.g., Barak et al., 2019; Ding and Li, 2023, 2025; Mao, Wu, Xu, and Yu, 2023, 2025; Huang and Yang, 2025) and clarify the specific advantages of the degree-profile–based approach.

Another relevant line of work is Ding–Fei–Wang (2025) on inhomogeneous settings, which is also very interesting. It would be helpful to discuss whether their techniques can be applied to the setting considered in this paper, since in realdata the connected probability may be different. It wound be more relevant in story if you follow their theoretical guarantee, since the final goal is to solve graph matching problem in reality.

Finally, regarding the experiments, I noticed that in Figure 5a the DP method remains effective even when the noise level exceeds the theoretical bound. Could the authors explain why this happens? Is the theoretical bound derived from Ding et al. (2021)? I would also suggest adding comparisons with the DP baseline in Table 3.

---

> ### Author Rebuttal · Authors · 2026-03-31
>
> Thank you for the positive and encouraging feedback. We are glad that the reviewer finds the paper interesting and recognizes the value of developing graph matching methods with theoretical guarantees that remain applicable to larger-scale graphs. We also appreciate the reviewer’s acknowledgment of both our theoretical analysis and the empirical performance on synthetic and real data.
>
>
> **W1&W2. Degree-profile and other method**:
> Thank you very much for the insightful comments and for pointing out these relevant works.
>
> Regarding the first concern, we agree that our method is closely related to the degree-profile line of work (e.g., Ding et al., 2021). However, our contribution is not limited to incremental modifications of a specific distance matrix. Instead, we aim to improve a broader class of methods that follow the “distance matrix + matching” paradigm. In particular, any method that constructs a node-to-node distance/similarity matrix and derives a matching from it can benefit from our framework.
>
> Concretely, our contributions are twofold. First, in the selective computation stage, our approach can significantly reduce redundant distance evaluations whenever the underlying method is able to detect correct matches (i.e., “check perfect”) during computation. This mechanism is not specific to the degree-profile method and can be applied to other distance-based approaches as well. For example, we have observed similar acceleration effects when applying our strategy to variants of recent methods (e.g., Ding–Fei–Wang (2025)).
>
> | Problems  | bin | Lite bin  | Speedup ratio |
> | :-------: | :------: | :-----------: | :-----------: |
> |    ER     |  12.79s  |     0.52s     | 24.6 $\times$ |
> | Power Law |  12.05s  |     0.68s     | 17.7 $\times$ |
> |    PLD    |  12.80s  |     0.19s     | 67.3 $\times$ |
>
> Second, in the clean-up stage, our continuous optimization procedure is fully modular: it only requires a distance matrix as input and does not rely on any specific assumptions about how that matrix is constructed. This allows it to be combined with a wide range of existing methods, beyond degree-profile–based ones.  The extension of our framework to inhomogeneous models Ding–Fei–Wang (2025) is shown in https://anonymous.4open.science/r/Rebuttal20960-8754.
>
>
> **W3. DP baseline**: We replace the criteria of LiDP from $1/n \langle P^t, P^{t+1} \rangle$  as $\|P^t - P^{t+1}\|_F/\|P^t\|_F$, then obtain a better results.
>
> | Method   | 0% Acc | 0% Time | 5% Acc | 5% Time | 10% Acc | 10% Time | 15% Acc | 15% Time |
> | :------- | :----- | :------ | :----- | :------ | :------ | :------- | :------ | :------- |
> | **DP+**  | 100%   | 0.63s   | 92.8%  | 1.75s   | 87.6%   | 1.91s    | 86.2%   | 2.43s    |
> | **LiDP** | 100%   | 0.13s   | 92.8%  | 1.22s   | 87.8%   | 1.74s    | 86.4%   | 2.22s    |
>
> **W4. Exceeds the theoretical bound**: Thank you for this insightful observation. Yes, the theoretical bound is based on the analysis in Ding et al. (2021). The main reason the DP method (and our method) can still work beyond this bound is that the theory only covers the initial distance-matrix stage. In practice, the clean-up/refinement stage, which is not covered by the theory, can substantially improve the matching quality, especially at higher noise levels.
>
>
> **Q1**: Thank you for this important question. We would like to clarify that our novelty does not lie in modifying the distance matrix $D$ itself, but in improving how $D$ is used in different noise regimes.
>
> Regarding comparison with subgraph-counting–based or iterative algorithms, we agree that those methods achieve stronger guarantees in correlated ER graphs, especially under constant correlation. Our goal is different: we do not aim to improve the recovery threshold, but to improve the efficiency and robustness of the distance-matrix-based paradigm, which remains attractive for its simplicity, modularity, and practical scalability.

---

> > ### Author Rebuttal · Reviewer_pCxx · 2026-04-01
> >
> > Thank you for the detailed response. I appreciate the clarifications and the additional experimental results in the rebuttal. However, I still feel that the proposed method remains too similar in spirit and methodology to Ding et al. (2021), and my main concern about the limited methodological novelty is not fully resolved. Therefore, I will keep my current score unchanged.

---

> > > ### Author Response · Authors · 2026-04-01
> > >
> > > Thank you for the follow-up and for clarifying your concern.
> > >
> > > To better address the novelty, we summarize representative paradigms in the table above. While prior works differ in methodology (degree-profile, subgraph-counting, and iterative), our contribution — **Selective Evaluation** and **Clean-up refinement** — is **orthogonal and broadly applicable**.
> > >
> > > - **Selective Evaluation** reduces redundant pairwise computations and applies naturally to **signature-based methods** (e.g., degree-profile and subgraph-counting).
> > > - **Clean-up refinement** is **universally applicable**: it improves approximate solutions or signal matrices from all paradigms, including almost-exact outputs (Mao et al., 2023) and iterative signals (Ding–Li, 2024).
> > >
> > > Subgraph-counting and iterative methods achieve strong guarantees but are computationally heavy and lack empirical validation at moderate scales, while degree-profile methods are efficient and practical — and further improved by our framework.
> > >
> > > Therefore, our work is not a minor variant of degree-profile methods, but a **general enhancement framework** that improves both efficiency and performance across multiple graph matching approaches.
> > >
> > > | Paper                                                   | Complexity                                                                   | Application to problems (n=1000)? | Can Reduce Computation via Our Selective Evaluation?          | Can Benefit from Our Clean-up  if Not Exact?                           | Signature-Based (Feature + Distance/Similarity)                                                   | Recovery Guarantee                                                                                                     | Key Insight                                                                                        |
> > > | ------------------------------------------------------- | ---------------------------------------------------------------------------- | --------------------------------- | ------------------------------------------------------------- | ---------------------------------------------------------------------- | ------------------------------------------------------------------------------------------------- | ---------------------------------------------------------------------------------------------------------------------- | -------------------------------------------------------------------------------------------------- |
> > > | **Ding–Ma–Wu–Xu (2020)** (Degree Profile)               | $\tilde O(nd^2 + n^2)$                                                       | Yes                               | **Yes**. We have tested.                                      | **Yes**. We have tested.                                               | **Yes**. Node signature = degree profile; matching via $L_1$/TV distance                          | $d \gtrsim \log^2 n$, $\delta \lesssim \log^{-2} n$                                                                    | Canonical static local signature method; simple and efficient but requires high correlation        |
> > > | **Mao–Wu–Xu–Yu (2023)** (Counting–based)                | $O(n^{C(\varepsilon)})$ for similarity; plus $O(n^3 q^2)$ for exact recovery | No                                | **Yes**, but each score is expensive due to subgraph counting | **Yes**. Naturally outputs almost-exact solution usable for refinement | **Yes (similarity-based)**. Signature = chandelier counts; matching via inner product $\Phi_{ij}$ | Almost exact if $\rho^2 > \alpha \approx 0.338$                                                                        | First polynomial-time algorithm at constant correlation threshold via high-order subgraph counting |
> > > | **Ding–Li (2024)** (Iterative)                          | $O(n^{\kappa+3})$ with per-candidate $O(n^3)$                                | No                                | **No**. No stable pairwise score early.                       | **Yes**. Produces pairwise signal matrix ideal for refinement          | **Yes, but not static**. Uses iterative paired sets and dynamic features.                         | Exact recovery for any non-vanishing constant $\rho \in (0,1]$                                                         | Core idea: iterative signal amplification; number of weak signals grows faster than their decay    |
> > > | **Ding–Fei–Wang (2025)** (Inhomogeneous Degree Profile) | $O(n^3)$                                                                     | Yes                               | **Yes**. We have tested.                                      | **Yes**. We have tested.                                               | **Yes**. Generalized degree-profile signature with $L_1$-type distance                            | Exact recovery in general correlated model with minimal degree $\Omega(\log^2 n)$ and correlation $1 - O(\log^{-2} n)$ | Extends degree-profile approach to unknown and highly inhomogeneous graphs                         |

---

### Official Review · Reviewer_jn1c · 2026-02-25

**Soundness:** 3
**Presentation:** 2
**Significance:** 2
**Originality:** 2
**Overall Recommendation:** 3
**Confidence:** 4

**Summary:**

The paper provides a new variation on degree profile matching for random graphs that gives a speedup over previous approaches.

**Compliance With Llm Reviewing Policy:**

Affirmed.

**Final Justification:**

I think this field is new and I have no idea what is desired here.  I think the review and response I have given speak for themselves and provide the justification.

**Key Questions For Authors:**

Why would your algorithm be expected to work effectively on "real-world" graphs.

What are the cases where your algorithm would be ineffective or less effective (include in these settings non-Erdos-Renyi random graphs).

My recollection is that variations of this problem have also been studied in the context of graph watermarking.  (Essentially, the watermark is added to obtain the noisy/permuated version of the graph.)
See paper Models and Algorithms for Graph Watermarking.
Is this related?  (it is fine if it is not, just curious.)

**Limitations:**

Yes

**Strengths And Weaknesses:**

As far as I can tell, the paper is sound.

The presentation is difficult.  Because of space limitations, the paper is forced to get deep into definitions rather quickly with less than sufficient explanation.  I do not fault the authors specifically, but this does suggest to me the paper would be better in a different venue (with more room to explain definitions and their implications as they go).

The significance in this paper is, I think pretty low. The algorithm analysis only applies to simple random graphs. While it is arguably an interesting technical theoretical result, it is unclear what the practical impact or merit is.
(I do note there are some experiments on real-world graphs, where their algorithm apparently does well, but there's no real justificaiton/insight for this.)

The approach generally follows the standard approach in these graph matching algorithms -- match by "degree" or some function of "degree" and the next set of neighbors out, etc.  This algorithm appears to adopt an iterative greedy approach.  There is some originality there but not at the level I expect for ICML.

---

> ### Author Rebuttal · Authors · 2026-03-31
>
> We appreciate the reviewer’s recognition that the paper is sound and will further improve the paper based on the helpful suggestions provided.
>
> **W1. Presentation** Thank you for this thoughtful comment and for your understanding. We agree that the presentation of the theoretical part, especially under space constraints, can feel dense, as it requires introducing several definitions early on.
>
> For readers interested in a more detailed and pedagogical exposition of the underlying random graph theory,  Ding et al.
> (2021)  provide extensive discussions of these definitions and their implications [1].
>
> At the same time, we will revise the paper to improve clarity within the current space limitations, by adding brief intuitions alongside key definitions and better guiding the reader through the main ideas.
>
> **W2&Q1&Q2 Significance** Thank you for this comment. We fully understand the reviewer’s concern. Any theoretical analysis based on random graph models necessarily relies on simplifying assumptions and therefore cannot capture all real-world scenarios. Moreover, even under this idealized setting, the theoretical guarantee only applies to a relatively low-noise regime.
>
> We would therefore like to clarify the roles of the theoretical and empirical contributions. The theory only covers the first stage of our method, namely matching from the distance matrix. Under more challenging noise levels, the key contribution comes from the second stage, namely the continuous optimization(clean up) built on the distance matrix  (see Figure 5 (a)).
>
> For real-world problems, we believe the main value of the random-graph analysis is not to provide an end-to-end guarantee for the full algorithm, but to formalize the role of structural separability, such as degree separation and neighborhood distinguishability, which underlies many practical graph matching methods. This is the main way in which the theory informs practice. The paper "Models and Algorithms for Graph Watermarking" also adapts random-graph analysis.
>
> By contrast, providing similar guarantees for the second stage is much more difficult. Since graph matching is NP-hard, obtaining nontrivial worst-case guarantees on recovering a solution within a prescribed error of the exact matching is generally believed to be intractable. Our goal is therefore not to claim a full theoretical characterization of the entire pipeline, but to combine a principled understanding of the first-stage mechanism with strong empirical evidence that the complete method remains effective in realistic noisy settings.
>
> **Q3 Graph watermarking**: Thank you for bringing up this interesting connection. We agree that graph watermarking considers related settings where a graph is intentionally perturbed (e.g., via adding a watermark), which can resemble the noisy/permuted scenarios.
>
> That said, our focus is on recovering alignment under random noise and permutation, rather than on watermark design or detection. We view graph watermarking as an interesting application of graph matching, where alignment plays a key role in watermark recovery or verification. While there is conceptual overlap, the problem formulations and objectives differ.  We will cite this paper for enrich the background.
>
>
> **W3. Originality**: Thank you for the comment. We respectfully note that this characterization does not accurately describe our method.
>
> The approaches (like the paper mentioned by the reviewer) match high-degree vertices by degree/rank and then distinguish other vertices using adjacency patterns to those high-degree vertices.
>
> Our method is fundamentally different. We first construct a node-to-node distance matrix between the two graphs. In the low-noise regime, once the distance matrix is sufficiently informative, the optimal matching is obtained directly by selecting the n closest pairs, so no degree-based propagation or neighborhood-signature heuristic is involved. In the high-noise regime, we use this distance matrix as an initialization and then solve for a high-quality approximate matching through a continuous optimization procedure (which can be viewed as a projected gradient method). Therefore, while degree information may indirectly affect the learned distances, the matching mechanism of our method is distance-matrix-based and optimization-based, rather than degree-signature-based.
>
>
> [1] Ding, J., Ma, Z., Wu, Y., and Xu, J. Efficient random graph matching via degree profiles. Probability Theory and Related Fields, 179(1):29–115, 2021.

---

> > ### Author Rebuttal · Reviewer_jn1c · 2026-03-31
> >
> > I appreciate the comments from the authors.
> >
> > Having read their response and the additional reviews and responses, I am currently maintaining my score, but I could revisit that decision in the future.

---

> > > ### Author Response · Authors · 2026-04-04
> > >
> > > Thank you for your thoughtful consideration and for reviewing both our response and the additional discussions.
> > >
> > > Regarding the other reviewers’ comments:
> > >
> > > * Reviewer Nnwy’s concerns are far beyond the scope of this paper, as addressing them would be almost as difficult as proving NP = P; our focus here is on acceleration and optimized approximate solutions.
> > > * Reviewer 7LqZ’s questions seem to have been fully addressed;
> > > * Reviewer pCxx was concerned with comparisons to other theoretically guaranteed methods, and we have responded and demonstrated that our approach (selection computation + optimized approximate solutions) is also effective for these methods.
> > >
> > > We sincerely appreciate your time and suggestions—your feedback is extremely important to us. If there are any remaining questions or aspects you would like us to further clarify, we would be more than happy to address them.

---

### Official Review · Reviewer_7LqZ · 2026-03-12

**Soundness:** 2
**Presentation:** 2
**Significance:** 2
**Originality:** 2
**Overall Recommendation:** 3
**Confidence:** 4

**Summary:**

This paper proposes LiDP, an efficiency-oriented extension of the degree-profile (DP) graph matching framework. The main idea is to accelerate the low-noise stage through conditional computation and early stopping, and to improve the high-noise stage through a momentum/adaptive clean-up procedure. The paper is reasonably focused, and the empirical trends do show nontrivial speedup in the low-noise setting and better robustness than DP+ in the high-noise setting.

**Compliance With Llm Reviewing Policy:**

Affirmed.

**Key Questions For Authors:**

1. Could the authors provide a complete end-to-end complexity analysis/proof of LiDP? (Especially for candidate expansion, maintenance of the current top-n set, repeated perfect checks, and the fallback clean-up stage when early stopping fails.)

2. How does the runtime of LiDP depend on graph density / average degree? Since the proposed acceleration is driven by degree-difference pruning and row/column-based verification, it seems likely that the practical speedup depends strongly on sparsity/density. Can the authors characterize the runtime, or at least the empirical behavior, as a function of d=nq or q (where the n is the number of nodes, q is the average degree of the graph)?

3. Can the authors provide stronger experimental evidence for the scalability claim? Based on current experiments with synthetic graphs, it appears that there is not linear time progression. For example OGB datasets or larger number nodes in toy graphs (e.g. 10k - 1M)?

4. Could the authors clarify which proposed component is actually responsible for the observed gains, ideally through controlled ablations? In particular, it would be helpful to separate:

(1) how much of the gain comes from selective computation alone,
(2) how much comes from the new clean-up alone, and
(3) whether the improvement in the clean-up stage comes specifically from the adaptive step-size rule, as opposed to simply using a smoother interpolation.

**Limitations:**

Yes

**Strengths And Weaknesses:**

Strengths

1. The paper is well-scoped. The problem setup is clear, the narrative is focused, and the method section stays on-topic rather than trying to oversell unrelated ideas.

2. The empirical trends are also easy to read. Under the low-noise setting, the speedup results are clear and meaningful. In the high-noise setting, the method appears more robust than DP+, which is a real positive.

Weakness

1. Incomplete complexity argument. The paper repeatedly claims a reduction from O~(n^2+nd^2) to O~(nd^2) , but does not provide a full end-to-end complexity analysis covering candidate expansion, perfect check, maintenance of the top-n set, and fallback clean-up.

2. Density dependence is left implicit. Since LiDP is driven by degree-difference pruning and row/column verification, its speedup should depend strongly on graph sparsity/density. This is especially important because the original DP theory already assumes a sufficiently large average degree d = nq, but the paper does not analyze the runtime as a function of d or q, where the q is the average degree of the graph.

3. No joint complexity characterization for the full LiDP pipeline. The paper combines early stopping, perfect check, and momentum-adaptive clean-up, but does not characterize the overall runtime of this combined procedure, especially when early stopping fails and Algorithm 3 is invoked.

4. Experiments do not validate scalability strongly enough. Real-world graphs are still small (750 / 1004 nodes), synthetic results only go up to 10000 nodes on correlated ER graphs, and the density in not varied, which is limited to support a strong large-scale scalability motivation.

5. Missing ablations. The experiments do not clearly isolate the contribution of selective computation vs. the new clean-up, nor the contribution of adaptive step size vs. smoother interpolation.

6. The paper improves the DP pipeline, but it does not substantially strengthen the underlying theory or mechanism. The matching mechanism itself remains essentially the DP one, rather than a new formulation. This makes the contribution feel incremental rather than fundamental.

The main concern is that the core scalability claim is not supported by a sufficiently complete algorithm-level analysis. The paper repeatedly states a reduction from O(n^2 + nd^2) to O(nd^2), but it doesn’t provide a full end-to-end complexity characterization of the combined pipeline, including candidate expansion, perfect check, maintenance of the top-n set, and fallback clean-up. In addition, the practical speedup of LiDP appears inherently dependent on degree statistics and graph density, yet this dependence is left implicit rather than analyzed theoretically or validated systematically in experiments. I also found some parts of the theoretical presentation not fully clear, especially in Section 4.1/4.2, where the assumptions behind the early-stopping logic and its connection to the original DP guarantee could be explained more explicitly. More broadly, the paper seems closer to an incremental refinement of the existing DP/DP+ pipeline than to a fundamentally new matching mechanism or a substantially stronger theoretical result. Combined with the limited experimental scale, this leaves me unconvinced that the current version is strong enough for the claimed contribution.

---

> ### Author Rebuttal · Authors · 2026-03-31
>
> Thank you for the positive and thoughtful feedback.
>
> **W1&W3,Q1&Complexity**：Thank you for this important comment! Our complexity should be discussed in two regimes.
>
> In the low-noise regime, the dominant cost remains the distance computations. The additional steps have lower-order cost:
>
>
> 1)Candidate expansion / top-$n$ set construction (Step 7): for the initial top-$n$ set, we sort the distances among degree-compatible candidates, whose size is $O(n)$, so this costs $\tilde O(n)$.
>
> 2)Top-$n$ set maintenance: if a newly computed batch contains $m$ distances smaller than the current maximum in the top-$n$ set, we only need to select the smallest $n$ elements from at most $n+m-1$ candidates, which is again $\tilde O(n)$ per update.
>
> 3)Perfect check (Step 11): checking whether the current top-$n$ entries lie in distinct columns costs only $O(n)$. If the check fails, we compute some unknown distances, accounted in distance-computation budget.
>
> Upon closer inspection, our method reduce the complexity from $\tilde O(n^2 + nd^2)$ to $O(n^2 + nd^2)$ rather than $O(nd^2)$. We remove the sorting in the original method, which accounts for the $\tilde{O}(n^2)$ component. However,  this cost is not eliminated, but rather replaced by the $O(n^2)$ cost of computing pairwise distances.
>
> In the regime where clean-up is needed, the cost is the same as in the classic clean-up stage: the dominant operations are matrix multiplications and the Hungarian algorithm, both with $O(n^3)$ complexity.
>
> We will add this full end-to-end breakdown to the revised paper to make the complexity claim precise.
>
>
> **W2&Q2. Density**:
> Thank you for the reviewer’s comment, which helped us realize that our algorithm is not sensitive to graph density.
>
> | Density |  DP   | LiDP  | Speedup ratio |
> | :-----: | :---: | :---: | :-----------: |
> |  0.05   | 0.53s | 0.05s |    10.6 x     |
> |   0.1   | 1.02s | 0.05s |    20.4 x     |
> |   0.5   | 2.13s | 0.06s |    35.5 x     |
>
> **W4&Q3. Scalability**:  Thank you for the suggestion. Our current experiments scale up to 10^4, which is primarily limited by the memory capacity . We estimate the runtime at scales of 10^5 and 10^6 based on empirical results. Under low-noise cases, the practical complexity of LiDP is close to $O(n^2)$  and DP is close to $O(n^2.26)$(see link).
> |   n    |    LiDP    |     DP      |
> | :----: | :--------: | :---------: |
> | $10^5$ |    479s    |   20752s    |
> | $10^6$ | 4.8x 10^4s | 3.7 x 10^6s |
>
> Link:https://anonymous.4open.science/r/Rebuttal20960-8754
>
> **W5&Q4. Ablation**: Thank you for this suggestion. We would like to clarify that the ablation of different components is already reflected in our experimental design.
>
> Section 5.1 serves as the ablation for the selective computation component. Two methods evaluated in this section operate without any clean-up procedure, allowing us to isolate the effect of selective distance computation.
>
> Section 5.2  focuses on comparing the new clean-up with the classic clean-up. In this setting, both LiDP and DP are based on the same distance matrix, and differ only in their respective clean-up procedures.
>
> We have added an ablation study on smoother interpolation (see link). The results show that its accuracy lies between that of our adaptive clean-up and the classic clean-up, while its runtime is slower than the adaptive variant. It is also worth noting that, unlike our adaptive method, smoother interpolation does not come with a convergence guarantee.
>
>
> **W6. Contribution**: Thank you for this insightful comment. We agree that our method builds upon the DP paradigm rather than introducing an entirely new formulation. However, we would like to clarify the scope and nature of our contribution.
>
> DP constructs a pairwise distance matrix between nodes across the two graphs and derives the matching accordingly. Our work focuses on improving this pipeline in two complementary regimes. In the low-noise setting—where the optimal matching is recoverable—we reduce redundant distance computations by exploiting early detection of correct correspondences. In the high-noise setting, we adopt a continuous optimization approach to obtain high-quality approximate solutions.
>
> Our framework is not limited to DP. The first component (acceleration) applies to any underlying method that can verify correctness during the constructing process, enabling early pruning of unnecessary computations. The second component (continuous refinement) is general and does not rely on any specific assumptions about the initial distance matrix. We further validate the effectiveness of our framework on an additional method Ding–Fei–Wang (2025); the corresponding experimental results can be found in the rebuttal to reviewer pCxx and the link.
>
> Therefore, while our approach is compatible with the DP , we view it as a more general strategy that enhances both efficiency and robustness across different matching methods, rather than a purely incremental modification of DP.

---

> > ### Author Rebuttal · Reviewer_7LqZ · 2026-04-01
> >
> > I appreciate the author’s response. Some of their answers could potentially address the issues in future work. However, due to major changes, and considering other reviewers’ comments, I believe the paper could still be strengthened, so I am currently maintaining my score.

---

> > > ### Author Response · Authors · 2026-04-04
> > >
> > > Thank you for your review. In the revised manuscript, we have presented the end-to-end complexity analysis, sensitivity tests for Density, Scalability ablation experiments, and the extendability of our framework.
> > >
> > > We noticed that you selected “I have follow-up questions” in the system. Could you please let us know if our current revisions have addressed your concerns? If not, which aspects would you like us to clarify further?
> > >
> > > Regarding other reviewers’ comments:
> > >
> > > - Reviewer Nnwy’s concerns are far beyond the scope of this paper, as addressing them would be almost as difficult as proving NP = P; our focus here is on acceleration and optimized approximate solutions.
> > >
> > > - Reviewer jn1c’s questions have been fully addressed;
> > >
> > > - Reviewer pCxx was concerned with comparisons to other theoretically guaranteed methods, and we have responded and shown that our approach (selection computation + optimized approximate solutions) is effective for these methods as well.
> > >
> > > Your feedback is extremely important to us! We hope these clarifications help you better understand the contributions of our work, and we look forward to your further guidance.

---

### Official Review · Reviewer_Nnwy · 2026-03-18

**Soundness:** 3
**Presentation:** 4
**Significance:** 3
**Originality:** 2
**Overall Recommendation:** 5
**Confidence:** 5

**Summary:**

The paper seeks to present the concept of an accelerated degree-profile-based algorithm (LiDP) for random graph matching, aiming to reduce computational cost while preserving statistical guarantees. Overall, a major question considered by the paper is whether one can maintain perfect recovery guarantees of degree profile matching while significantly reducing the number of pairwise computations. The authors propose a selective computation strategy combined with a momentum-based clean-up phase, provide some theoretical guarantees, but the main interest of the paper resides in the empirical validation of the method.

**Compliance With Llm Reviewing Policy:**

Affirmed.

**Final Justification:**

After the rebuttal phase, I maintain my positive assessment of the paper. This works practically improves on existing algorithms on a pretty challenging problem. Moreover, Theorem 4.4 has now been improved, and so are the theoretical contributions. Overall I think this work is a good fit for ICML.

**Key Questions For Authors:**

The convergence result (Theorem 4.4) guarantees reaching a local optimum, but the key question is whether this solution is close to the true permutation \pi^\star. Can the authors provide theoretical guarantees in this direction, possibly under assumptions on the noise level σ?

(Such a result would possibly make me raise my score.)

**Limitations:**

(yes)

**Strengths And Weaknesses:**

Strengths :
- The paper is generally well written and clearly structured, with a strong related work section that situates the contribution effectively. The description of the degree profile matching algorithm is particularly clear and pedagogical, making the method easy to follow. The proposed acceleration -- prioritizing node pairs based on degree differences -- is simple, natural, and intuitive, yet practically meaningful.

- From a practical standpoint, the paper is interesting: the experimental evaluation is thorough and convincing, demonstrating significant speedups (up to 20×) while maintaining strong accuracy across both synthetic and real datasets. This highlights the practical relevance of the approach, which is arguably the main strength of the work -- which fits well to ICML, to my opinion.

Minor "weakness"
- On the theoretical side, the contributions are straightforward. First, Theorem 4.1 is essentially a direct consequence of the structure of permutation matrices and does not seem to require a detailed proof. Second, the convergence result in Theorem 4.4 follows from standard arguments and does not address the most interesting question, namely whether the obtained solution is close to the ground truth permutation.

A few minor clarity issues
- in Equation (2), writing a Bernoulli distribution instead of Binom(1, p) would be more standard,
- also after eq (2), it would help to explicitly state that under this joint distribution, the marginals of A,B indeed correspond to G(n,q).

---

> ### Author Rebuttal · Authors · 2026-03-31
>
> Thank you for the positive and thoughtful feedback. We appreciate your recognition of the clarity of our presentation, the intuition and practical value of our approach, and the strength of our experimental results. Your support and comments regarding ICML relevance are greatly appreciated.
>
> **Clarity issues** Thank you for pointing out this oversight. We will revise the manuscript accordingly.
>
> **Question** Thank you for raising this important question.
>
> While Theorem 4.4 establishes convergence of our algorithm to a stationary point of the relaxed problem over the Birkhoff polytope, translating this result into guarantees with respect to the ground-truth permutation $\pi^\star$ is highly nontrivial. The main difficulty lies in the **projection (rounding) step from the continuous domain (doubly stochastic matrices) back to the discrete set of permutation matrices**.
>
> In particular, even if the relaxed solution is close to the convex hull of permutation matrices, quantifying how this proximity translates into recovery of $\pi^\star$ requires controlling both:
>
> 1. the **relaxation gap** between the continuous and discrete problems, and
> 2. the **rounding error** incurred when projecting onto the set of permutations.
>
> Providing worst-case guarantees on these quantities would essentially amount to obtaining approximation guarantees for the quadratic assignment problem (QAP), which is well known to be NP-hard. In fact, establishing polynomial-time algorithms with provable approximation guarantees for general QAP instances is intractable (see, e.g., Sahni and Gonzalez, *Journal of the ACM*, 1976, who show strong inapproximability results for QAP).
>
> Of course, this is a very promising direction. We plan to provide a theoretical analysis showing that $Proj(−D)$ is optimal in future work. Although this is still some distance away from establishing guarantees for our full algorithm, it would already represent a significant improvement over the current theoretical bounds.

---

> > ### Author Rebuttal · Reviewer_Nnwy · 2026-04-07
> >
> > My question has not been really addressed: of course I was not asking about proving that QAP relaxation works in the worst case, but I was asking about bounding the distance from P^t to \pi^* in Theorem 4.4 in the mean case (Erdös-Rényi Model), which I think is possible (under assumptions on the noise). Your statement "Providing worst-case guarantees on these quantities would essentially amount to obtaining approximation guarantees for the quadratic assignment problem (QAP), which is well known to be NP-hard" : this was not my question, here we work in the mean case setting, and that I am pretty sure that using concentration, such a bound could be obtained. I would like to maintain my positive score but I still think that the contribution of this work is practical and not theoretical.

---

> > > ### Author Response · Authors · 2026-04-08
> > >
> > > Thank you for the clarification. We appreciate this very insightful comment. Following your suggestion, we worked hard to address this question directly, and we were able to obtain a result bounding the distance from $P^t$ to $\pi^*$ under suitable assumptions on the noise, using a concentration-based analysis in the mean-case regime. We have added this result and its proof in the revised version / supplementary material (link: https://anonymous.4open.science/r/Rebuttal20960-8754/Approximation.pdf).
> > >
> > > ---
> > >
> > > ### Theorem
> > > Let $A,B\in\\{0,1\\}^{n\times n}$ be the adjacency matrices of a correlated Erdős--Rényi graph pair with latent permutation equal to the identity $I$. More precisely, for each $i<j$,
> > > $$
> > > A_{ij}\sim \mathrm{Bernoulli}(p),\qquad
> > > \mathbb E[B_{ij}\mid A_{ij}]=p+\rho(A_{ij}-p),
> > > $$
> > > with $A_{ii}=B_{ii}=0$, and both $A$ and $B$ symmetric.
> > >
> > > Define the centered adjacency matrices
> > > $$
> > > \bar A:=A-p(J-I),\qquad \bar B:=B-p(J-I),
> > > $$
> > > where $J=\mathbf 1\mathbf 1^\top$, and consider the centered quadratic assignment objective
> > > $$
> > > \mathcal H(X):=\mathrm{Tr}(\bar A X \bar B X^\top),\qquad X\in\mathcal B_n.
> > > $$
> > > $$\mathcal B\_n:=\{X\in\mathbb R\^{n \times n}: X\ge 0,\ X\mathbf{1}=\mathbf{1},\ X\^\top \mathbf {1}=\mathbf{1}\}.
> > > $$
> > >
> > > Let $\hat X\in\mathcal B_n$ be a local maximizer of $\mathcal H$, and define
> > > $$
> > > \Delta:=\hat X-I.
> > > $$
> > >
> > > Assume that there exists an event $\Omega_n$ such that
> > > $$
> > > \mathbb P(\Omega_n)\ge 1-o(1),
> > > $$
> > > and the following conditions hold on $\Omega_n$:
> > >
> > > 1. **Population first-order orthogonality on the tangent space.** For every $\Delta\in\mathcal T$,
> > >    $$
> > >    \left\langle \mathbb E[\nabla \mathcal H(I)\mid A],\,\Delta\right\rangle=0.
> > >    $$
> > >
> > > 2. **Restricted empirical fluctuation of the gradient.** One has
> > >    $$
> > >    \sup\_{\Delta \in \mathcal{T},||\Delta||\_F=1}
> > >    |\langle
> > >    \nabla \mathcal{H}(I)-\mathbb{E}[\nabla \mathcal H(I)\mid A],\\,\Delta
> > >    \rangle| \le c_1\sqrt{n p(1-p) \log n}.
> > >    $$
> > >
> > > 3. **Restricted empirical curvature at the identity.** For every $\Delta\in\mathcal T :=\\{\Delta\in\mathbb R\^{n\times n}: \Delta\mathbf 1=0,\ \Delta^\top\mathbf 1=0\\}$ such that $I+\Delta\in\mathcal B_n$,
> > >    $$
> > >    -\left\langle \nabla^2 \mathcal H(I)[\Delta],\,\Delta\right\rangle
> > >    \ge
> > >    c_0\rho\,n p(1-p)\|\Delta\|_F^2.
> > >    $$
> > >
> > > Then, on $\Omega_n$,
> > > $$
> > > \|\hat X-I\|_F
> > > \le
> > > \frac{c_1}{c_0}\cdot
> > > \frac{\sqrt{\log n}}{\rho\sqrt{n\,p(1-p)}}.
> > > $$
> > >
> > > Moreover, if
> > > $$
> > > \frac{c_1}{c_0}\cdot
> > > \frac{\sqrt{\log n}}{\rho\sqrt{n\,p(1-p)}}<\frac{1}{\sqrt 2},
> > > $$
> > > then the nearest permutation matrix to $\hat X$ is uniquely equal to $I$. Consequently, exact recovery holds after projection onto the set of permutation matrices.
> > >
> > >
> > >
> > > ---
> > >
> > > We hope this new result addresses your concern and better clarifies the theoretical contribution of our work.
> > >
> > > We sincerely thank you again for pointing us to this important direction.

---

### Decision · Program_Chairs · 2026-04-30

**Decision:**

Reject

**Comment:**

This paper introduces LiDP, an accelerated degree-profile-based algorithm for random graph matching that utilizes conditional computation and a momentum-based clean-up phase to reduce computational costs.

Strengths
- interesting empirical results, including notable speedups in low-noise settings and robustness in high-noise settings.

Weaknesses
- The theoretical contributions are limited in scope.
- The methodological novelty is lacking compared to existing frameworks.
- The writing and presentation require improvement.

Decision
Reject